# Effects of nutrient supply on leaf stoichiometry and relative growth rate of three stoloniferous alien plants

**Dong-Wei Yu**[1], **Su-Juan Duan**[1], **Xiao- Chao Zhang**[2], **Da-Qiu Yin**[3], **Shi-Jun Wang**[3], **Jin-Song Chen**[1]*, **Ning-Fei Lei**[2]*

**1** College of Life Science, Sichuan Normal University, Chengdu, China, **2** College of Ecology and Environment, Chengdu University of Technology, Chengdu, China, **3** China Huaneng Group Co., Ltd, Beijing, China

* cjs74@163.com (J-SC); leiningfei@cdut.cn (N-FL)

**Data Availability Statement:** All relevant data are within the paper and its Supporting Information files.

## Abstract

Different nutrient supply brings about changes in leaf stoichiometry, which may affect growth rate and primary production of plants. Invasion of alien plants is a severe threat to biodiversity and ecosystem worldwide. A pot experiment was conducted by using three stoloniferous alien plants *Wedelia trilobata*, *Alternanther philoxeroides* and *Hydrocotyle vulgaris* to investigate effects of nutrient supply on their leaf stoichiometry and relative growth rate. Different nitrogen or phosphorus supply was applied in the experiment (N1:1 mmol L$^{-1}$, N2:4 mmol L$^{-1}$, and N3:8 mmol L$^{-1}$, P1:0.15 mmol L$^{-1}$, P2:0.6 mmol L$^{-1}$ and P3:1.2 mmol L$^{-1}$). Nitrogen and phosphorus concentrations in leaves of the three alien plants significantly increased with increase of nitrogen supply. With increase of phosphorus supply, nitrogen or phosphorus concentration of leaf was complex among the three alien plants. N:P ratio in leaf of the three alien plants subjected to different levels of nutrient supply was various. A positive correlation between relative growth rate and N:P ratio of the leaf is observed in *W. trilobata* and *A. philoxeroides* suffering from N-limitation. A similar pattern was not observed in *Hydrocotyle vulgaris*. We tentatively concluded that correlations between relative growth rate and N: P ratio of the leaf could be affected by species as well as nutrient supply. It is suggested that human activities, invasive history, local abundance of species *et al* maybe play an important role in the invasion of alien plants as well as relative growth rate.

## Introduction

N:P ratio is a critical indicator of nutrient limitation (N vs P) in the terrestrial ecosystem [1]. Leaf stoichiometry can reflect nutrient allocation strategy, growth strategy and expanding ability of invasive plants [2]. Plant stoichiometry refers to balance and ratio of C, N, P in its issue or organ [3]. Primary production and litter decomposition of plants were significantly affected by C: N: P ratio of the leaf as well as community dynamics and nutrient cycling [4–9]. In addition, C: N: P ratio of the leaf was related to plant adaptation to specific environments [10,11].

**Funding:** This work was supported by the Specialized Fund for the Post-Disaster Reconstruction and Heritage Protection in Sichuan Province(No.5132202019000128) and Study and Application on Technological System of Ecological Repair for Surface Destroyed by Hydropower Station Engineering Located in Middle Reaches of Yaluzangbu River (SKLGP2021Z018). The funders had no role in study design, data collection and analysis, decision to publish, or preparation of the manuscript.

**Competing interests:** The authors have declared that no competing interests exist.

N:P ratio of the leaf may be useful to assess nitrogen versus phosphorus limitation to primary production in terrestrial ecosystems [12–15].

Different nitrogen and phosphorus supply bring about changes of leaf stoichiometry, and these changes are various among different plants [16]. For example, with increase of nitrogen availability, phosphorus concentration in the leaf of desert grass *Seriphidium korovinii* and two alpine grasses significantly increased [17,18]. Meanwhile, an opposite pattern was observed in the phosphorus concentration of leaf [19–22]. With increase of nitrogen availability, nitrogen concentration in the leaf significantly increased [23,24]. In central Brazil, the similar pattern was not observed in four plants (*Caryocar Brasiliense*, *Qualea parviflora*, *Schefflera macrocarpa* and *Ouratea hexasperma*) grown on a dystrophic soil [25]. Nitrogen concentration in the leaf of *Arabidopsis thaliana* and desert grasses was not significantly affected by phosphorus supply [20,21]. With increase of phosphorus supply, nitrogen concentration in the leaf of the oak *Quercus acutissima* and shrub *Rhus typhina* significantly decreased [21,24]. With increase of phosphorus supply, phosphorus concentration in leaves of the alpine grassland significantly increased [18]. With increase of phosphorus supply, phosphorus concentration in the leaf of *S. korovinii* was various [20]. So, the effects of nitrogen and phosphorus supply on leaf stoichiometry of plants are still controversial.

With a lower N:P ratio, faster-growing organisms (such as microbes, zooplankton, arthropod and insect) often present greater tissue phosphorus concentration than slower-growing ones [26]. Relative growth rate of *Betula pendula* and *Arabidopsis thaliana* grown in P-limited conditions was negatively correlated with the N:P ratio of leaf. At the same time, a positive correlation between the N:P ratio of leaf and relative growth rate was detected in the two species suffering from N-limited conditions [16,27–29]. Relative growth rate of mangroves was not significantly affected by the N:P ratio of leaf [30]. More studies are needed to investigate the correlation between N:P ratios of leaves and relative growth rate, especially in vascular plants [27,30,31].

The invasion of alien plants severely threatens biodiversity and ecosystem worldwide [32–34]. Alien plants often accumulate more nitrogen and/or phosphorus than the co-occurring native ones[35,36]. The possible driving factor for the successful invasion of alien plants is their greater nutrient capture capacity than native ones[37,38]. However, the clear relationship between nutrient absorption capacity of alien plants and their expanding ability was not established in other studies[39,40]. Further studies are thus needed to clarify the correlations between leaf stoichiometry of alien plants and their growth performance.

Stoloniferous alien plants *Wedelia trilobata*, *Alternanther philoxeroides* and *Hydrocotyle vulgaris* originate from Europe and/or America respectively. The three alien plants are widely distributed in the south of the Yangtze River, China [41,42]. A pot experiment was conducted (1) to evaluate the effects of nutrient supply on leaf nitrogen and phosphorus concentrations of the three alien plants; (2) to explore the correlation between leaf N: P ratio of the three alien plants and their relative growth rate. Finally, our study aims to understand the potential influence of leaf stoichiometry on invasion of the three alien plants.

## Materials and methods

### Species

*Wedelia trilobata*, *Alternanther philoxeroides* and *Hydrocotyle vulgaris* are perennial stoloniferous herbs. Axillary buds on the vertical stem of *W. trilobata* may grow out to form stolon. Ramets with opposite leaves take root at the node of stolon, which forms a network aboveground [43]. *W. trilobata* is distributed predominantly in tropical and subtropical regions such as Asia and South America [42].

The stolon of *Alternanther philoxeroides* usually takes root when in contact with moist substratum, forming a network of stolon [44]. *A. philoxeroides* is distributed in many areas of the world such as Australia, the United States, and China [45,46].

Each node along the stolon of *Hydrocotyle vulgaris* has the potential to produce an independent ramet that consists of a simple leaf and adventitious roots [47]. *H. vulgaris* is widely distributed in tropical and temperate regions such as Southeast Asia, Europe and North America.

In 2018, 10 original plants of *W. trilobata* were collected from Zhejiang Province, China. 20 original plants of *A. philoxeroides* and *H. vulgaris* were collected in Sichuan Province, China. All original plants were planted in a greenhouse located at Sichuan Normal University. The average temperature is about $25 \pm 5°C$ in the greenhouse.

## Experimental design

In September 2019, stolon internodes of the three alien plants without leaves were chosen. The stolon internodes were about 3-5cm length in size. Balance was used to measure their fresh weight. According to the ratio of dry to fresh weight, we could obtain initial biomass of each stolon internode [19]. The stolon internodes were respectively grown in a plastic pot (12 cm in upper edge diameter, 9 cm in bottom diameter and 10.5 cm in height) filled with perlite. Same volume of nutrient solution was added to each plastic pot every two days.

Three levels of nitrogen supply (N1: 1mmol $L^{-1}$, N2:4 mmol $L^{-1}$, and N3: 8 mmol $L^{-1}$ nitrogen, added as $NH_4NO_3$) and three levels of phosphorus supply (P1:0.15 mmol $L^{-1}$, P2:0.6 mmol $L^{-1}$, and P3:1.2 mmol $L^{-1}$ phosphorus, added $KH_2PO_4$ and $NaH_2PO_4$ as 1:1 ratio) were applied in the experiment (Table 1). In addition, macro- and microelements (such as 0.75 mM $K_2SO_4$, 0.65 mM $MgSO_4$, 1 μM $MnSO_4$, 0.1 μM $CuSO_4$, 1 μM $ZnSO_4$, 0.035 μM $Na_2MoO_4$, 0.1 mM Fe-EDTA, 0.01 mM $H_3BO_3$, and 2 mM $CaC_{l2}$) were added to nutrient solution [16]. The pH of nutrient solution was adjusted to 5.8. The experiment was carried out in a greenhouse with 16-h light/8-h dark photoperiod and $25 \pm 5°C$. Each treatment was replicated six times. The experiment lasted for 60 days. To avoid potential effects of the micro-environment, the pots were randomly rearranged every two days.

At the end of the experiment, fully expanded and mature leaves were selected from regenerated clonal fragments and dried to constant weight. The leaf samples were powdered and sifted through 100-mesh. Mortar Autoanalyzer 3 system (Seal analytical, Germany) was used to measure the nitrogen and phosphorus concentrations of the leaf [48].

Finally, whole regenerated clonal fragments were harvested and dried to constant weight. Balance was used to determine their biomass.

**Table 1. Nitrogen and phosphorus supply used in the experiment.**

| Level of nitrogen supply | Level of phosphorus supply | Nitrogen supply (mmol $L^{-1}$) | Phosphorus supply (mmol $L^{-1}$) | N:P (mol:mol) |
|---|---|---|---|---|
| Low nitrogen (N1) | Low phosphorus (P1) | 1 | 0.15 | 6.7 |
| | Intermediate phosphorus (P2) | 1 | 0.6 | 1.7 |
| | High phosphorus (P3) | 1 | 1.2 | 0.8 |
| Intermediate nitrogen (N2) | Low phosphorus (P1) | 4 | 0.15 | 26.7 |
| | Intermediate phosphorus (P2) | 4 | 0.6 | 6.7 |
| | High phosphorus (P3) | 4 | 1.2 | 3.3 |
| High nitrogen (N3) | Low phosphorus (P1) | 8 | 0.15 | 53.3 |
| | Intermediate phosphorus (P2) | 8 | 0.6 | 13.3 |
| | High phosphorus (P3) | 8 | 1.2 | 6.7 |

The relative growth rate was calculated according to following formula:

$$\text{Relative growth rate} = \frac{\text{Ln}(M_t) - \text{Ln}(M_0)}{t} \qquad (1)$$

In the experiment, $M_0$ is the initial biomass of the stolon internode, $M_t$ is the final biomass of regenerated clonal fragment, and t is duration of the experiment.

We estimate the balance of plant homeostasis by stoichiometric regulation coefficient $H$.

The stoichiometric regulation coefficient H was calculated according to the following formula:

$$H = \frac{\text{Log}_{10}(x)}{\text{Log}_{10}(y) - \text{Log}_{10}(c)} \qquad (2)$$

In the experiment, x is N:P or C: N or C:P ratio of nutrition, y is N:P or C: N or C:P ratio of leaf, c is a constant[49].1/H is a slope value and it should be between 0 and 1. The classifications of H values are as follows: H>4(homeostasis), 4>H>2(weak homeostasis), 2>H>1.33 (weakly sensitive state), 1.33>H(sensitive state). But it is often understood as absolute homeostasis when the equation fitting the data is insignificant.

## Statistical analyses

Two-way analysis of variance was performed to assess the effects of nitrogen, phosphorus supply and their interaction on leaf stoichiometry of the three alien plants. Linear regression was performed to explore the correlation between relative growth rate of the three alien plants and their leaf stoichiometry respectively. All analyses were conducted with SPSS 20.0 software (SPSS Inc).

## Results

Nitrogen and phosphorus concentrations in the leaf of *W. trilobata* and *H. vulgari*s were significantly affected by nitrogen supply as well as interaction between nitrogen and phosphorus supply (Tables 2 and 3). Phosphorus concentration in the leaf of *W. trilobata* was significantly affected by phosphorus supply (Table 2). Nitrogen and phosphorus concentrations in the leaf of *A. philoxeroides* was significantly affected by nitrogen supply, phosphorus supply as well as interaction between nitrogen and phosphorus supply (Table 4).

With increase of nitrogen supply, nitrogen and phosphorus concentrations in the leaf of *W. trilobata* significantly increased (Figs 1A and 2A). Nitrogen concentration in the leaf of *H. vulgaris* significantly increased (Fig 1B). Phosphorus concentration in the leaf of *H. vulgaris* significantly increased (P2 or P3). Phosphorus concentration in the leaf of H. vulgaris was

**Table 2. Effects (*F* value) of nitrogen and phosphorus supply and their interaction on the relative growth rate of *W. trilobata*, nitrogen and phosphorus concentrations, N: P ratio in its leaf.**

| Factors | Relative growth rate (g day $^{-1}$) | Nitrogen concentration (mg g $^{-1}$) | Phosphorus concentration (mg g $^{-1}$) | N:P ratio |
|---|---|---|---|---|
| **Nitrogen supply** | 26.16*** | 13.83*** | 57.10*** | 31.49*** |
| **phosphorus supply** | 0.76 | 2.36 | 99.44*** | 30.25*** |
| **Interaction between nitrogen and phosphorus supply** | 2.36* | 5.01** | 4.60** | 25.71*** |

**P < 0.01

*P < 0.05, $^{ns}$P ≥ 0.05.

**Table 3. Effects (*F* value) of nitrogen and phosphorus supply and their interaction on relative growth rate of *H. vulgaris*, nitrogen and phosphorus concentrations, N: P ratio in its leaf.**

| Factors | Relative growth rate (g day$^{-1}$) | Nitrogen concentration (mg g$^{-1}$) | Phosphorus concentration (mg g$^{-1}$) | N:P ratio |
|---|---|---|---|---|
| Nitrogen supply | 7.93** | 281.40*** | 30.43*** | 348.79*** |
| Phosphorus supply | 12.08*** | 2.22 | 2.48 | 25.08*** |
| Interaction between nitrogen and phosphorus supply | 0.61 | 9.16*** | 9.07*** | 0.72 |

**P < 0.01

*P < 0.05, $^{ns}$P ≥ 0.05.

significantly higher in the treatment (N2P1) than in the treatment (N1P1). Then, phosphorus concentration in the leaf of *H. vulgaris* was significantly lower in the treatment (N3P1) than in the treatment (N2P1) (Fig 2B). Nitrogen concentration in the leaf of *A. philoxeroides* was significantly higher in treatment (N2P2) than in treatment (N1P2). Then, nitrogen concentration in the leaf of *A. philoxeroides* was significantly lower in the treatment (N3P2) than in the treatment (N2P2) (Fig 1C). Phosphorus concentration in the leaf of *A. philoxeroides* significantly increased (Fig 2C).

With increase of phosphorus supply, phosphorus concentration in the leaf of *W. trilobata* significantly increased (Fig 2A). With increase of phosphorus supply, nitrogen concentration in the leaf significantly decreased when *A. philoxeroides* was subjected to nitrogen supply (N1 or N2). Nitrogen concentration in the leaf of *A. philoxeroide* was significantly lower in the treatment (N3P2) than in the treatment (N3P1). Then, nitrogen concentration in the leaf of *A. philoxeroide* was significantly higher in the treatment (N3P3) than in the treatment (N3P2) (Fig 1C). Phosphorus concentration in the leaf of *A. philoxeroide* was significantly lower in the treatment (N3P1) than in the treatment (N3P2). Then, phosphorus concentration in the leaf of *A. philoxeroide* was significantly higher in the treatment (N3P2) than in the treatment (N3P3). With increase of phosphorus supply, phosphorus concentration in the leaf significantly decreased when *A. philoxeroides* was subjected to nitrogen supply (N2) (Fig 2C).

N:P ratio in leaf of *W. trilobata* or *A. philoxeroides* was significantly affected by nitrogen, phosphorus supply and interaction between nitrogen and phosphorus supply (Tables 2 and 4). With increase of nitrogen supply, N:P ratio in leaf significantly increased when *W. trilobata* was subjected to phosphorus supply (P1 or P2). N:P ratio of leaf was significantly lower in the treatment (N1P3) than in the treatment (N2P3). Then, N:P ratio of leaf was significantly higher in the treatment (N2P3) than in the treatment (N3P3). With increase of phosphorus supply, N:P ratio in leaf significantly decreased when *W. trilobata* was subjected to nitrogen supply (N1 or N3). N:P ratio of leaf was significantly lower in the treatment (N2P1) than in the treatment

**Table 4. Effects (*F* value) of nitrogen and phosphorus supply and their interaction on the relative growth rate of *A. philoxeroides*, nitrogen and phosphorus concentrations, N: P ratio in its leaf.**

| Factors | Relative growth rate (g day$^{-1}$) | Nitrogen concentration (mg g$^{-1}$) | Phosphorus concentration (mg g$^{-1}$) | N:P ratio |
|---|---|---|---|---|
| Nitrogen supply | 2.00 | 135.46*** | 90.03*** | 62.04*** |
| Phosphorus supply | 1.40 | 4.17* | 7.12** | 13.76*** |
| Interaction between nitrogen and phosphorus supply | 1.37 | 16.09*** | 20.30*** | 7.62*** |

**P < 0.01

*P < 0.05, $^{ns}$P ≥ 0.05.

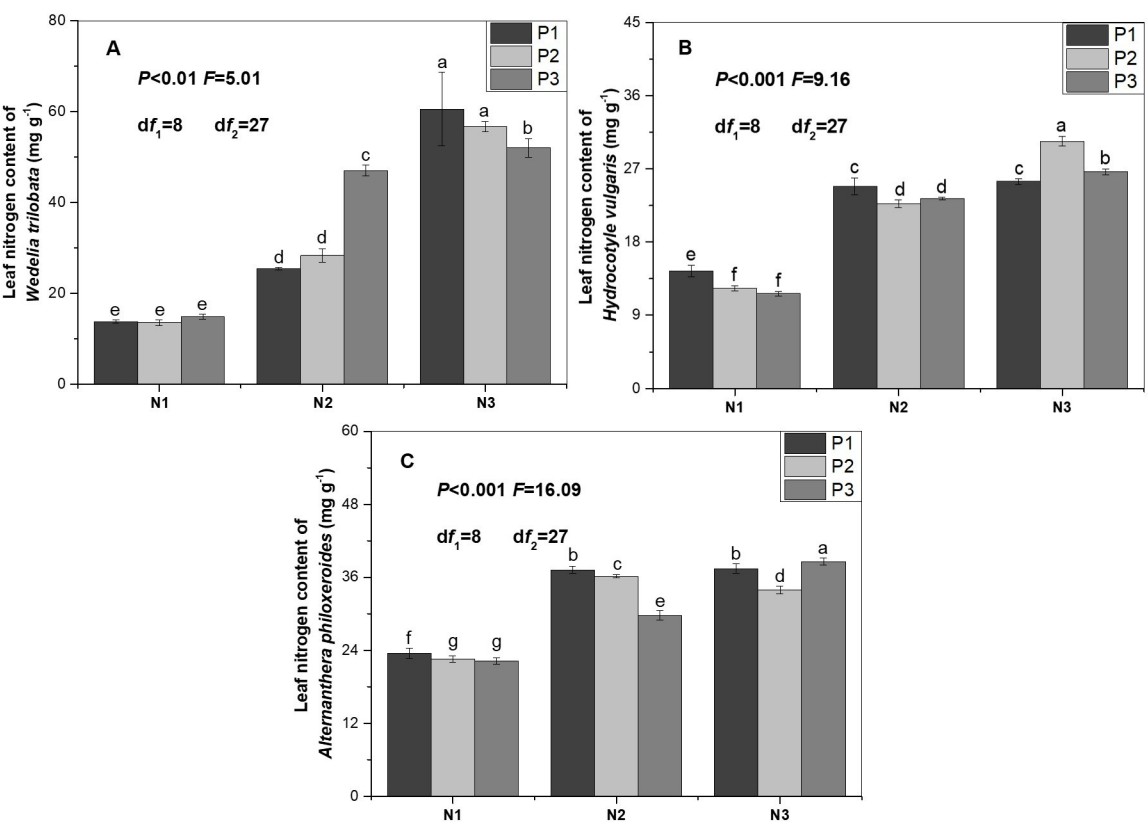

**Fig 1. Nitrogen concentration in leaf of the three alien plants was subjected to different nutrient supplies.** N1: Low nitrogen supply (1 mmol L$^{-1}$); N2: Intermediate nitrogen supply (4 mmol L$^{-1}$); N3: High nitrogen supply (8 mmol L$^{-1}$); P1: Low phosphorus supply (0.15 mmol L$^{-1}$); P2: Intermediate phosphorus supply (0.6 mmol L$^{-1}$); P3: High phosphorus supply (1.2 mmol L$^{-1}$). Values are means ± SE. Bars with different lowercase letters denote significant differences ($P < 0.05$).

(N2P2). Then, N:P ratio of leaf was significantly higher in the treatment (N2P2) than in the treatment (N2P3) (Fig 3A).

With increase of nitrogen supply, N:P ratio in leaf significantly increased when *H. vulgaris* was subjected to the same level of phosphorus supply (P1, P2, P3). With increase of phosphorus supply, N:P ratio in leaf significantly decreased when *H. vulgaris* was subjected to nitrogen supply (N2 or N3). N:P ratio of leaf was significantly higher in the treatment (N1P2) than in the treatment (N1P1). Then, N:P ratio of leaf was significantly lower in the treatment (N1P3) than in the treatment (N1P2) (Fig 3B).

With increase of nitrogen supply, N:P ratio in leaf significantly decreased when *A. philoxeroides* was subjected to phosphorus supply (P3). N:P ratio of leaf was significantly lower in the treatment (N1P1) than in the treatment (N2P1). Then, N:P ratio of leaf was significantly higher in the treatment (N2P1) than in the treatment (N3P1). N:P ratio of leaf was significantly lower in the treatment (N1P2) than in the treatment (N2P2). Then, N:P ratio of leaf was significantly higher in the treatment (N2P2) than in the treatment (N3P2). With increase of phosphorus supply, N:P ratio in leaf significantly decreased when *A. philoxeroides* was subjected to nitrogen supply (N2). N:P ratio of leaf was significantly lower in the treatment (N1P2) than in the treatment (N1P1). Then, N:P ratio of leaf was significantly higher in the treatment (N1P3) than in the treatment (N1P2). N:P ratio of leaf was significantly lower in the treatment (N3P2) than in the treatment (N3P1). Then, N:P ratio of leaf was significantly higher in the treatment (N3P3) than in the treatment (N3P2) (Fig 3C).

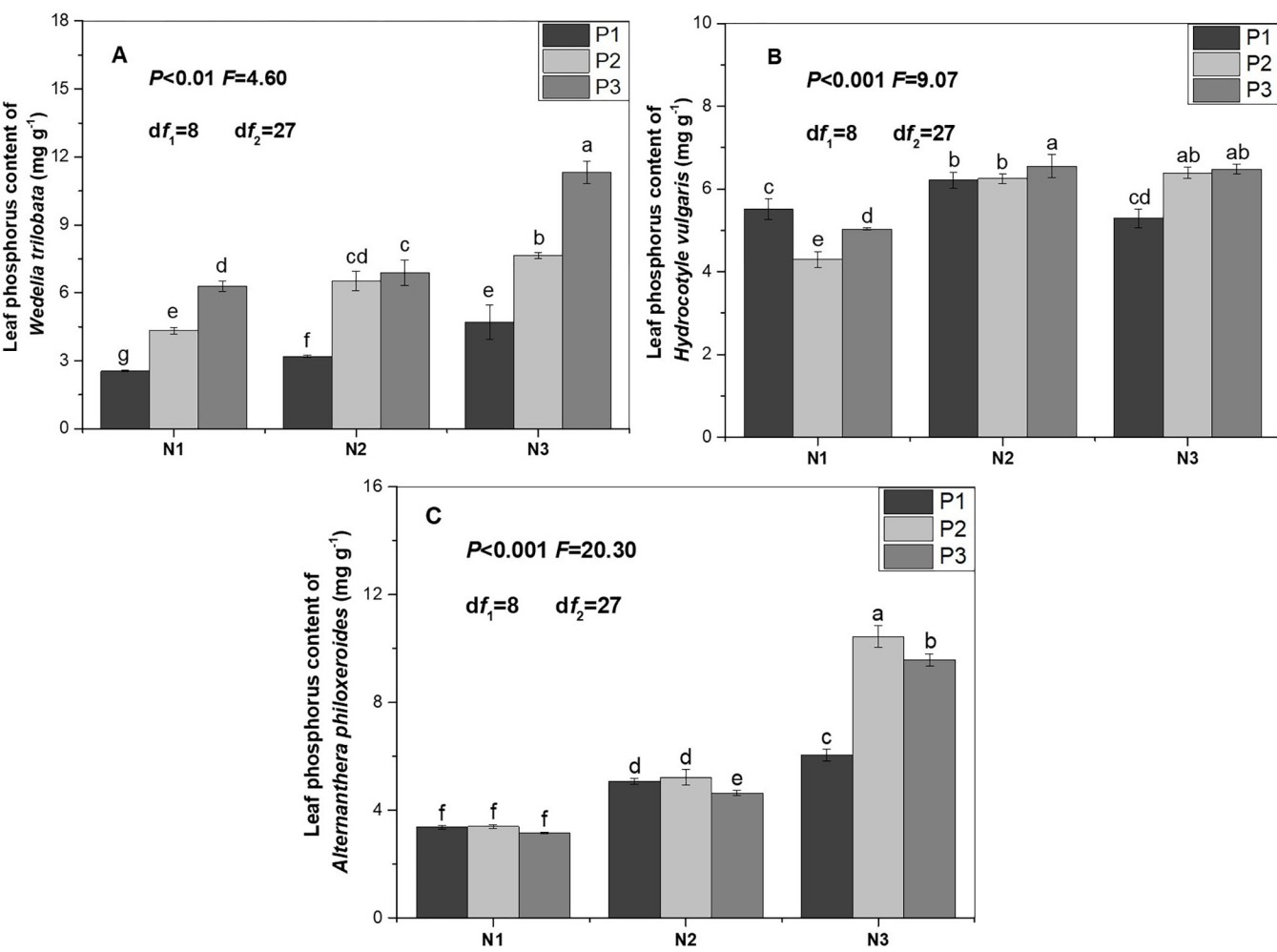

**Fig 2. Phosphorus concentration in leaf of the three alien plants was subjected to different nutrient supplies.** N1: Low nitrogen supply (1 mmol L$^{-1}$); N2: Intermediate nitrogen supply (4 mmol L$^{-1}$); N3: High nitrogen supply (8 mmol L$^{-1}$); P1: Low phosphorus supply (0.15 mmol L$^{-1}$); P2: Intermediate phosphorus supply (0.6 mmol L$^{-1}$); P3: High phosphorus supply (1.2 mmol L$^{-1}$). Values are means ± SE. Bars with different lowercase letters denote significant differences< 0.05).

$H_{(N:P)}$ of *W. trilobata* and *H. vulgaris* was 2.78 and 6.25 respectively. The relationship between N:P ratio of leaf and N:P ratio in nutrition solution was not established in *A. philoxeroide* (Table 5). N:P ratio in leaf of *W. trilobata* and *A. philoxeroides* was positively correlated with their relative growth rate (Fig 4A and 4C). However, a similar correlation was not observed in *H. vulgaris* (Fig 4B).

## Discussion

Nitrogen or phosphorus concentration in leaf of the three alien plants significantly increased with increase of nitrogen supply, which is consistent with previous studies [50,51]. On the one hand, nitrogen addition increased the nitrogen concentration of leaf [52–55]. On the other hand, nitrogen addition exerted a positive effect on phosphatase activity, which enhances phosphorus acquisition of plants grown in high phosphorus availability of soil [56–58].

Nitrogen concentration in the leaf of *W. trilobata* and *H. vulgaris* was not significantly affected by phosphorus supply, which is consistent with the previous study [59]. With increase

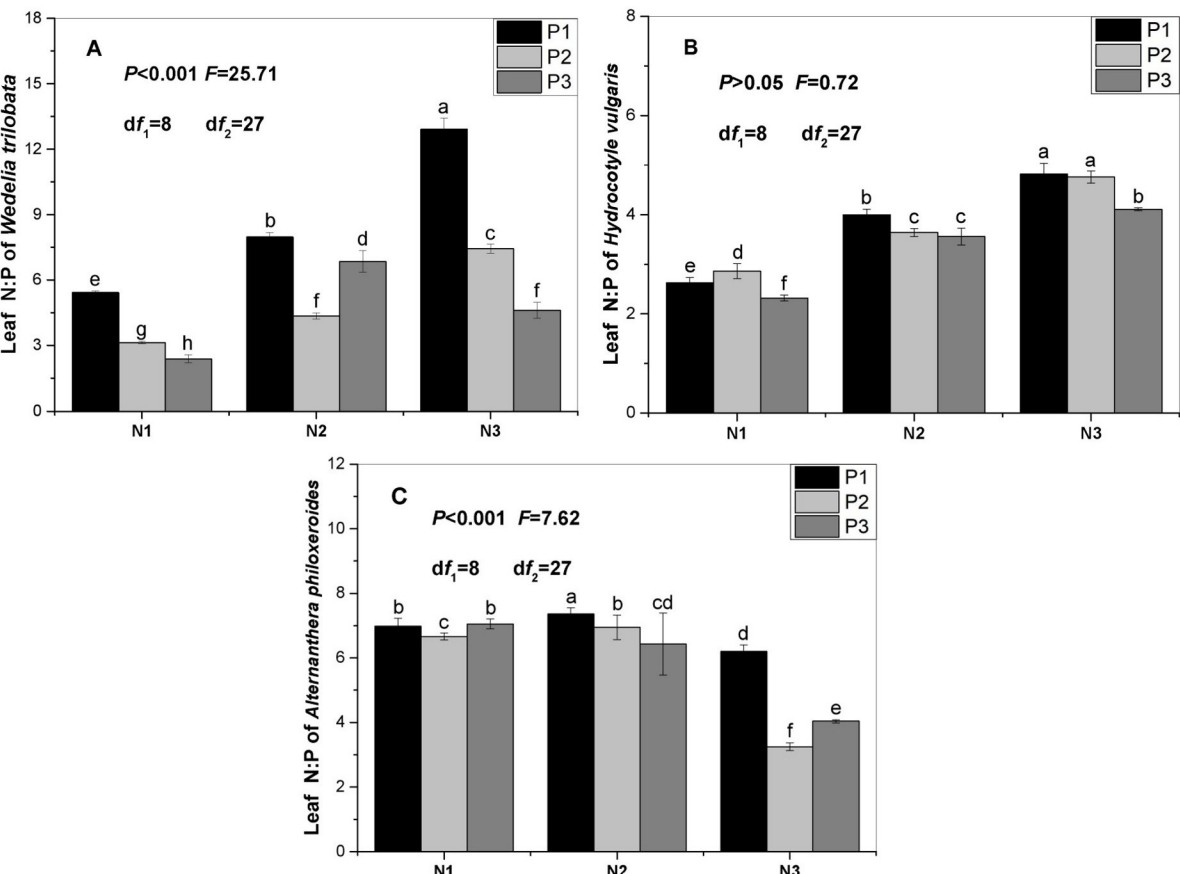

**Fig 3. P ratio in leaf of the three alien plants subjected to different nutrient supplies. N:** N1: Low nitrogen supply (1 mmol L$^{-1}$); N2: Intermediate nitrogen supply (4 mmol L$^{-1}$); N3: High nitrogen supply (8 mmol L$^{-1}$); P1: Low phosphorus supply (0.15 mmol L$^{-1}$); P2: Intermediate phosphorus supply (0.6 mmol L$^{-1}$); P3: High phosphorus supply (1.2 mmol L$^{-1}$). Values are means ± SE. Bars with different lowercase letters denote significant differences ($P < 0.05$).

of phosphorus supply, a tendency of decreasing was observed in the nitrogen concentration of leaf when *A. philoxeroides* was subjected to nitrogen supply (N1 or N2). The possible explanation is that phosphorus addition incurred the use of internal nitrogen in issue or organ of plant [18]. As the same time, the tendency of decreasing first and then increasing was observed in the nitrogen concentration of leaf when *A. philoxeroides* was subjected to nitrogen supply (N3). The possible explanation is that efficiency and proficiency of N resorption significantly

**Table 5. The homeostatic regulation coefficient (H) and parameters calculated from the linear regression for leaf ecological stoichiometry (N:P) of three alien plants with stolon and their congenerics.**

| Species | Rate | 1/H | R$^2$ | H | Stoichiometric homeostasis | Relative growth rate |
|---|---|---|---|---|---|---|
| *W. trilobata* | N:P | 0.36*** | 0.83 | 2.78*** | Weak homeostasis | 0.065±0.008 *** |
| *H. vulgaris* | N:P | 0.16** | 0.67 | 6.25** | homeostasis | 0.077±0.007 *** |
| *A.philoxeroides* | N:P | -0.04 | 0.04 | -25.00 | absolute homeostasis | 0.064±0.009 ** |

***P<0.001

**P< 0.01

*P<0.05.

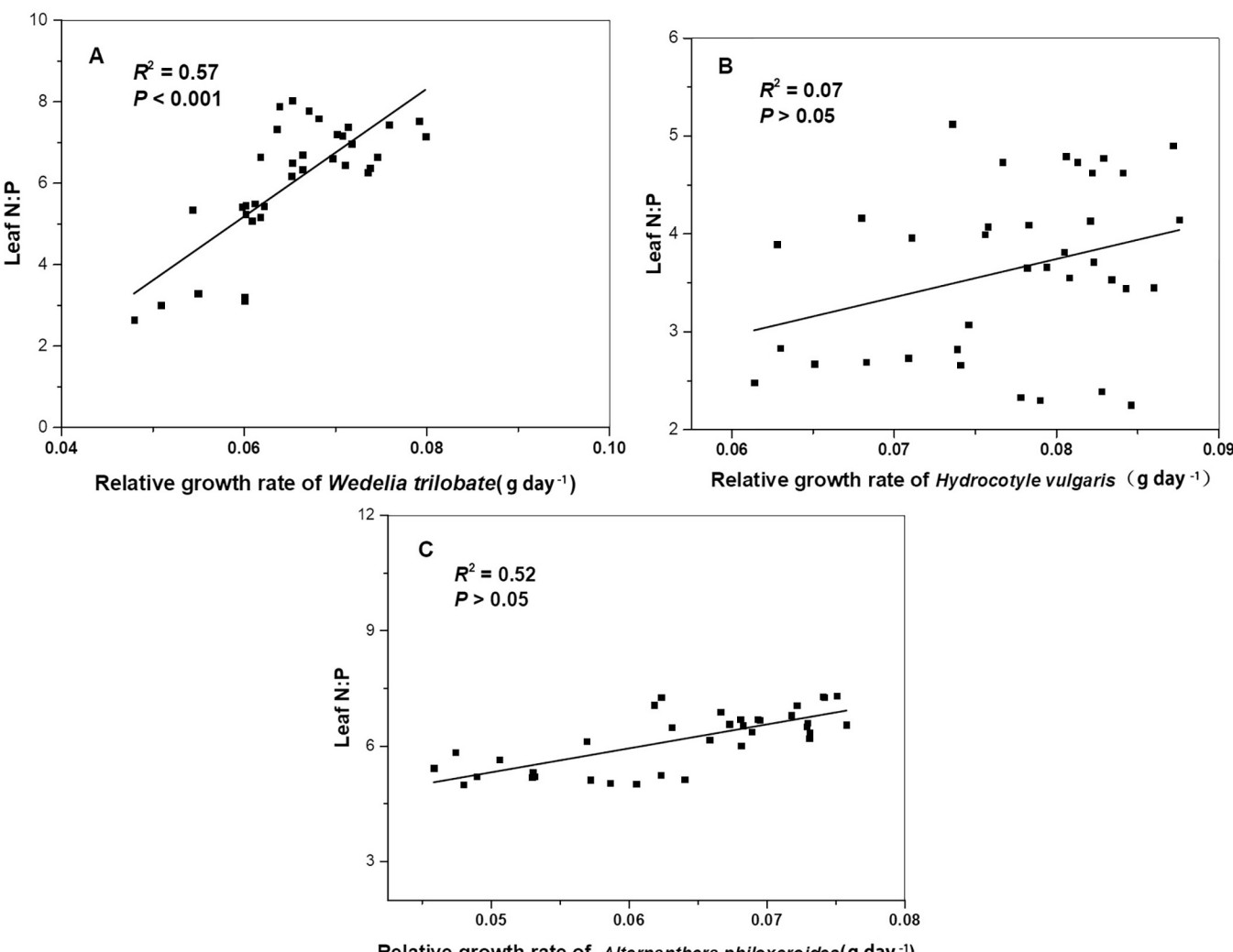

**Fig 4. Relationship between the relative growth rate of the three alien plants and N:P ratios of their leaf.**

decreased in plants subjected to high nitrogen supply [18,59]. Phosphorus addition increased phosphorus concentration in leaves of alpine grassland [18]. Responses of phosphorus concentration in the three alien plants to phosphorus addition were complex, which is similar to the previous study [20].

Compared with those in previous studies, N: P ratio (<14) in leaf of the three alien plants was lower [27,60]. The possible explanation is that the three alien plants are suffering from N-limitation in the experiment [16,29,61]. N:P ratio in leaf was various when the three alien plants were subjected to different levels of nutrient supply. So, more studies are needed to explore the relationship between N:P ratio and nutrient supply.

In the experiment, relative growth rates of *W. trilobata*, *A. philoxeroides* and *H. vulgaris* were 0.065±0.008, 0.064±0.009 and 0.077±0.007 respectively. Based on their $H_{(N:P)}$, *W. trilobata*, *A. philoxeroides* and *H. vulgaris* were classified as weak homeostasis, absolute homeostasis and homeostasis respectively [62]. Stoichiometric homeostasis presented a positive influence on relative growth rate of plants growing in different nutrient availability [49]. Species dominance in Inner Mongolia grassland was positively correlated with their

stoichiometric homeostasis [63,64]. So, further studies are needed to explore the relationship between stoichiometric homeostasis and growth performance of the plant.

Relationship between relative growth rate of plants and the leaf N: P ratio of its leaf was affected by nutrient limitation [16,28]. With nitrogen addition, a positive correlation between relative growth rate and N: P ratio of the leaf was observed in *Arabidopsis thaliana* suffering from N-limitation [16]. Negative correlation between relative growth rate and N: P ratio of the leaf was observed in *N. tangutorum* [49]. Correlations between relative growth rate and N: P ratio of leaf were different among *W. trilobata*, *A. philoxeroides* and *H. vulgaris*. We tentatively concluded that correlations between relative growth rate and N: P ratio of the leaf could be affected by species as well as nutrient supply.

*W. trilobata* is distributed in Guangxi Zhuang Autonomous Region, Guangdong Province, Fujian Province, Hainan Province, China. *A. philoxeroides* is distributed in Guangxi Zhuang Autonomous Region, Guangdong Province, Fujian Province, Sichuan Province, Hebei Province, Yunnan Province, Chongqing, Hunan Province, Jiangxi Province, Zhejiang Province, Hubei Province, Henan Province, Anhui Province, Jiangsu Province, Shandong Province, China. *H. vulgaris* is distributed in Yunnan province, China (https://doi.org/10.15468/39omei). In the non-native range, the relative growth rate of invasive plants (*Agropyron cristatum and Bromus inermis*) was higher than native plants (*Elymus canadensis* and *Pascopyrum smithii*) [65]. In addition, a high relative growth rate could partly explain the successful invasion of alien plants [66]. A similar pattern was not observed in the experiment. It is suggested that human activities, invasive history, local abundance of species *et al* maybe play an important role in invasion of alien plants as well as relative growth rate.

## Supporting information

**S1 Data.**
(XLSX)

## Author Contributions

**Conceptualization:** Dong-Wei Yu, Jin-Song Chen, Ning-Fei Lei.

**Data curation:** Dong-Wei Yu, Xiao- Chao Zhang, Da-Qiu Yin, Shi-Jun Wang.

**Formal analysis:** Dong-Wei Yu, Su-Juan Duan, Shi-Jun Wang.

**Funding acquisition:** Su-Juan Duan, Shi-Jun Wang.

**Methodology:** Dong-Wei Yu, Xiao- Chao Zhang.

**Software:** Xiao- Chao Zhang.

**Writing – original draft:** Dong-Wei Yu.

**Writing – review & editing:** Jin-Song Chen, Ning-Fei Lei.

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
