## [Decision Letter · Decision Letter 0]

11 Oct 2022

PONE-D-22-22959Effects of nutrient supply on leaf stoichiometry and relative growth rate of three stoloniferous alien plantsPLOS ONE

Dear Dr. Chen,

Thank you for submitting your manuscript to PLOS ONE. After careful consideration, we feel that it has merit but does not fully meet PLOS ONE’s publication criteria as it currently stands. Therefore, we invite you to submit a revised version of the manuscript that addresses the points raised during the review process.

We look forward to receiving your revised manuscript.

Kind regards,

Xiao Guo, Ph.D.

Academic Editor

PLOS ONE

Journal Requirements:

Additional Editor Comments:

The authors compared the leaf stoichiometry and relative growth rate of three stoloniferous alien plants in response to nutrient supply. The topic has potential and would be of interest to ecologists and Invasion biologists.

I agree with both reviewers and recommend minor revision.

Reviewers' comments:

Reviewer's Responses to Questions

**Comments to the Author**

1. Is the manuscript technically sound, and do the data support the conclusions?

Reviewer #1: Yes

Reviewer #2: Yes

2. Has the statistical analysis been performed appropriately and rigorously? 

Reviewer #1: Yes

Reviewer #2: Yes

3. Have the authors made all data underlying the findings in their manuscript fully available?

Reviewer #1: Yes

Reviewer #2: Yes

4. Is the manuscript presented in an intelligible fashion and written in standard English?

Reviewer #1: Yes

Reviewer #2: Yes

5. Review Comments to the Author

Reviewer #1: In the paper, the authors addressed the effects of nutrient supply on their leaf stoichiometry and relative growth rate. As I know, the researches on the correlationship bewteen leaf stoichiometry of alien clonal plants are scarce. In this sense, this manuscript is a very good paper. Additionally, the experiment is also well designed and results are also very interesting. However, there are a couple of points that will need to be added or improved.

Line 2, “change” should be “changes”.

Line 7-8， “leaf” should be “leaves”

Line 10，“level” should be “levels”。

Line 10 and Line 12，“correlationship” should be “correlation”。

Line 14， “may be play an important role….” should be “may play an important role….”or “maybe play an important role….”。

Line 20， “plant” should be “plants”。

Line 29， “similar pattern” should be “the similar pattern”.

Line 37， “such as microbe” should be “such as microbes”。

Line 49， “bewteen” should be “between”。

Line 81， “was applied in the experiment” should be “were applied in the experiment”。

Line 84， “pH of nutrient” should be “The pH of nutrient”。

Line 104-106， “The classification of H values is” should be “classifications of H values are”

Line 106-107, “it often is” should be “it is often”.

Line 106-107, “fit” should be “fitting”.

Line 146， Fig 1 “supply ” should be “supplies”。

Line 207-209，“decreased in plant subjected high” should be “decreased in plants subjected to high”.

Line 212, “Comparing with” should be “Compared with”.

Line 214, “N:P ratio in leaf of was” should be “N:P ratio in leaf was”.

Line 215-216, “more studies are need” should be “more studies are needed”.

Line 215-216, “between N:P ration” should be “between N:P ratio”.

Line 220-221, “presented positive influence” should be “presented a positive influence”.

Line 224, “growth rate of plant and its leaf N:P ratio” should be “growth rate of plants and the N:P ratio”.

Line 230, “nutrientsupply” should be “nutrients supply”.

Line 234, “Jinagxi Province” should be “Jiangxi”.

Line 240， “ may be play” should be “ may play” or “maybe play”.

Reviewer #2: The manuscript described leaf stoichiometry of three alien plant species subjected to different N/P availability. The results provided evidence on relationship between leaf stoichiometry and relative growth rate. However，correlations between relative growth rate and N: P ratio of the leaf could be affected by species as well as nutrient supply. Overall, the article is well organized. The study may be helpful to understand effects of leaf stoichiometry on invasion of the three alien plants. In addition, some modifications are still needed to improve quality of the manuscript.If possible, the authors might polish the language in the manuscript with the help of a native English speaker.

Abstract:

Line6: ”N1:1 mmol L-1” . The expression is not clear on what is added.

Line 7-9: Throughout the manuscript, author refer to nutrient ‘contents’. I think it would be more accurate to refer to nutrient ‘concentrations’ in the plant tissue.

Line 14: Local abundance of species was not studied and the grammar of this sentence has error.

Introduction:

Line 19-20: There are many other mechanisms by which plants adapt to nitrogen and phosphorus deficient soils. It would be useful to briefly discuss these before focusing on leaf stoichiometry.

Line 24-25：In introduction, leaf stoichiometry would be changed by different nitrogen and phosphorus supply. The pattern is inconsistent among different plants. It is suggested that author may reorganize the sentence to express clearly.

Line 48-49: The sentence is difficult to understand. It is suggested that author rewrite the sentence to express clearly.

Materials and methods:

Further detail on the three species is not enough. It is suggested that more details on the three species are needed, for example, their invasive potential in China.

Tables and Figures:

The units for relative growth rate (i.e. g g–1 d–1) should be g day–1 (growth per day)?

Language editing:

Line 2, change “change” into “changes”.

Line 7-8, change “leaf” into “leaves”

Line 10, change “level” into “levels”

Line 40, change “……, positive correlation between……” into “……, a positive correlation between……”

Line 43, change “……between N:P ratio of leaf and…..” into “…..between N:P ratios of leaves and…..”

Line 45, change “Invasion of alien plants is a severe threat to biodiversity and ecosystem worldwide.” into “The invasion of alien plants severely threatens biodiversity and ecosystem worldwide.”

Line 49，change “bewteen” into “between”

Line 65, change “stolen” into “stolon”.

Line 67, change “Each node along stolon of…..” into “Each node along the stolon of…..”.

Line 81，change “was applied in the experiment” into “were applied in the experiment”

Line 158, change “diffdifferences” into “differences”.

Line 158, change “Table 2 and 4” into “Tables 2 and 4”

Line 234, change “Jinagxi Province” into “Jiangxi”.

6. PLOS authors have the option to publish the peer review history of their article (what does this mean?). If published, this will include your full peer review and any attached files.

Reviewer #1: No

Reviewer #2: No

---

## [Author Response · Author response to Decision Letter 0]

30 Oct 2022

Dear Editors:

Thank for comments from two anonymous reviewers. Those comments are very valuable for revising and improving our manuscript. We have revised the manuscript according to comments. All changes have been marked in red. The responses to the reviewer’s comments are as following:

Reviewer#1:

1. Line 2, “change” should be “changes”. We change “change” into “changes”

2. Line 7-8， “leaf” should be “leaves”. We change “leaf” into “leaves”

3. Line 10，“level” should be “levels”. We change “level” into “levels”

4. Line 10 and Line 12，“correlationship” should be “correlation”. We change “correlationship” into “correlation”

5. Line 14， “may be play an important role….” should be “may play an important role….”or “maybe play an important role….”.We change “may be play an important role….” Into “maybe play an important role….”

6. Line 20， “plant” should be “plants”. We change “plant” into “plants”

7. Line 29， “similar pattern” should be “the similar pattern”. We change “similar pattern” into “the similar pattern”

8. Line 37， “such as microbe” should be “such as microbes”. We change “such as microbe” into “such as microbes”

9. Line 49， “bewteen” should be “between”. We change “bewteen” into “between”

10. Line 81， “was applied in the experiment” should be “were applied in the experiment”. We change “was applied in the experiment” into “were applied in the experiment”

11. Line 84， “pH of nutrient” should be “The pH of nutrient”. We change “pH of nutrient” into “The pH of nutrient”

12. Line 104-106， “The classification of H values is” should be “classifications of H values are”. We change “The classification of H values is” into “classifications of H values are”

13. Line 106-107, “it often is” should be “it is often”. We change “it often is” into “it is often”

14. Line 106-107, “fit” should be “fitting”. We change “fit” into “fitting”

15. Line 146， Fig 1 “supply ” should be “supplies”. We change “supply” into “supplies”

16. Line 207-209，“decreased in plant subjected high” should be “decreased in plants subjected to high”. We change “decreased in plant subjected high” into “decreased in plants subjected to high”

17. Line 212, “Comparing with” should be “Compared with”. We change “Comparing with” into “Compared with”

18. Line 214, “N:P ratio in leaf of was” should be “N:P ratio in leaf was”. We change “N:P ratio in leaf of was” into “N:P ratio in leaf was”.

19. Line 215-216, “more studies are need” should be “more studies are needed”. We change “more studies are need” into “more studies are needed”

20. Line 215-216, “between N:P ration” should be “between N:P ratio”. We change “between N:P ration” into “between N:P ratio”

21. Line 220-221, “presented positive influence” should be “presented a positive influence”. We change “presented positive influence” into “presented a positive influence”

22. Line 224, “growth rate of plant and its leaf N:P ratio” should be “growth rate of plants and the N:P ratio”. We change “growth rate of plant and its leaf N:P ratio” into “growth rate of plants and the N:P ratio”

23. Line 230, “nutrient supply” should be “nutrients supply”. We change “nutrient supply” into “nutrients supply”

24. Line 234, “Jinagxi Province” should be “Jiangxi”. We change “Jinagxi Province” into “Jiangxi”

25. Line 240， “ may be play” should be “ may play” or “maybe play”. We change “may be play” into “maybe play” 

Reviewer#2:

1. Line6: “N1:1 mmol L-1”. The expression is not clear on what is added. Details on the nitrogen or phosphorus supply were mentioned in the sentence (line83-85).

2. Line 7-9: Throughout the manuscript, author refer to nutrient ‘contents’. I think it would be more accurate to refer to nutrient ‘concentrations’ in the plant tissue. We change ‘contents’ into ‘concentrations’

3. Line 14: Local abundance of species was not studied and the grammar of this sentence has error. Significant correlation between invasion ability and relative growth rate was not observed in the three alien plants. So, it is suggested that local abundance of species, human activities, invasive history et al may play an important role in invasion of alien plants as well as relative growth rate. 

4. Line 19-20: There are many other mechanisms by which plants adapt to nitrogen and phosphorus deficient soils. It would be useful to briefly discuss these before focusing on leaf stoichiometry. N:P ratio is a critical indicator of nutrient limitation (N vs P) in the terrestrial ecosystem. Leaf stoichiometry can reflect nutrient allocation strategy, growth strategy and expanding ability of invasive plants.

5. Line 24-25：In introduction, leaf stoichiometry would be changed by different nitrogen and phosphorus supply. The pattern is inconsistent among different plants. It is suggested that author may reorganize the sentence to express clearly. We rewritten the sentence into “Different nitrogen and phosphorus supply bring about changes of leaf stoichiometry, and these changes are various among different plants”. 

6. Line 48-49: The sentence is difficult to understand. It is suggested that author rewrite the sentence to express clearly. We rewritten the sentence into “However, the clear relationship between nutrient absorption capacity of alien plants and their expanding ability was not established in other studies”.

7. In the section of materials and methods, further detail on the three species is not enough. It is suggested that more details on the three species. More information on the three alien plants was supplemented in line54-56 and line237-242.

8. In the section of tables and figures, the units for relative growth rate (i.e. g g–1 d–1) should be g day–1 (growth per day)? We have made correction according to the comments.

9. Line 2, change “change” into “changes”. We change “change” into “changes”

10. Line 7-8, change “leaf” into “leaves”. We change “leaf” into “leaves”

11. Line 10, change “level” into “levels”. We change “level” into “levels”.

12. Line 40, change “……, positive correlation between……” into “……, a positive correlation between……”. We change “……, positive correlation between……” into “……, a positive correlation between……”.

13. Line 43, change “……between N:P ratio of leaf and…..” into “…..between N:P ratios of leaves and…..”. We change “……between N:P ratio of leaf and…..” into “…..between N:P ratios of leaves and…..”.

14. Line 45, change “Invasion of alien plants is a severe threat to biodiversity and ecosystem worldwide.” into “The invasion of alien plants severely threatens biodiversity and ecosystem worldwide.” We change “Invasion of alien plants is a severe threat to biodiversity and ecosystem worldwide.” into “The invasion of alien plants severely threatens biodiversity and ecosystem worldwide.”

15. Line 49，change “bewteen” into “between”. We change “bewteen” into “between”.

16. Line 65, change “stolen” into “stolon”. We change “stolen” into “stolon”.

17. Line 67, change “Each node along stolon of…..” into “Each node along the stolon of…..”. We change “Each node along stolon of…..” into “Each node along the stolon of…..”.

18. Line 81，change “was applied in the experiment” into “were applied in the experiment” We change “was applied in the experiment” into “were applied in the experiment”

19. Line 158, change “diffdifferences” into “differences”. We change “diffdifferences” into “differences”

20. Line 158, change “Table 2 and 4” into “Tables 2 and 4”. We change “Table 2 and 4” into “Tables 2 and 4”.

21. Line 234, change “Jinagxi Province” into “Jiangxi”. We change “Jinagxi Province” into “Jiangxi”.

We greatly appreciate help from you and two anonymous referees again. We hope that the revised manuscript may be acceptable for publication. We look forward to hearing from you.

Sincerely

 Dongwei Yu

---

## [Decision Letter · Decision Letter 1]

15 Nov 2022

PONE-D-22-22959R1Effects of nutrient supply on leaf stoichiometry and relative growth rate of three stoloniferous alien plantsPLOS ONE

Dear Dr. Chen,

Thank you for submitting your manuscript to PLOS ONE. After careful consideration, we feel that it has merit but does not fully meet PLOS ONE’s publication criteria as it currently stands. Therefore, we invite you to submit a revised version of the manuscript that addresses the points raised during the review process.

We look forward to receiving your revised manuscript.

Kind regards,

Xiao Guo, Ph.D.

Academic Editor

PLOS ONE

Journal Requirements:

Reviewers' comments:

Reviewer's Responses to Questions

**Comments to the Author**

1. If the authors have adequately addressed your comments raised in a previous round of review and you feel that this manuscript is now acceptable for publication, you may indicate that here to bypass the “Comments to the Author” section, enter your conflict of interest statement in the “Confidential to Editor” section, and submit your "Accept" recommendation.

Reviewer #1: All comments have been addressed

Reviewer #2: All comments have been addressed

2. Is the manuscript technically sound, and do the data support the conclusions?

Reviewer #1: Yes

Reviewer #2: Yes

3. Has the statistical analysis been performed appropriately and rigorously? 

Reviewer #1: Yes

Reviewer #2: Yes

4. Have the authors made all data underlying the findings in their manuscript fully available?

Reviewer #1: Yes

Reviewer #2: Yes

5. Is the manuscript presented in an intelligible fashion and written in standard English?

Reviewer #1: Yes

Reviewer #2: Yes

6. Review Comments to the Author

Reviewer #1: Manuscript PONE-D-22-22959R1 has been substantially improved after revision.

The authors have satisfactorily responded to most of the comments in my previous report.

Reviewer #2: All the comments have been addressed.The manuscript is relatively well-organized. I evaluated the manuscript and suggest for its publication

7. PLOS authors have the option to publish the peer review history of their article (what does this mean?). If published, this will include your full peer review and any attached files.

Reviewer #1: No

Reviewer #2: No

---

## [Author Response · Author response to Decision Letter 1]

18 Nov 2022

Dear Editors:

Thank for comments from two anonymous reviewers. Those comments are very valuable for revising and improving our manuscript. We have revised the manuscript according to comments. All changes have been marked in red. The responses to the reviewer’s comments are as following:

Reviewer#1:

1. Line 2, “change” should be “changes”. We change “change” into “changes”

2. Line 7-8， “leaf” should be “leaves”. We change “leaf” into “leaves”

3. Line 10，“level” should be “levels”. We change “level” into “levels”

4. Line 10 and Line 12，“correlationship” should be “correlation”. We change “correlationship” into “correlation”

5. Line 14， “may be play an important role….” should be “may play an important role….”or “maybe play an important role….”.We change “may be play an important role….” Into “maybe play an important role….”

6. Line 20， “plant” should be “plants”. We change “plant” into “plants”

7. Line 29， “similar pattern” should be “the similar pattern”. We change “similar pattern” into “the similar pattern”

8. Line 37， “such as microbe” should be “such as microbes”. We change “such as microbe” into “such as microbes”

9. Line 49， “bewteen” should be “between”. We change “bewteen” into “between”

10. Line 81， “was applied in the experiment” should be “were applied in the experiment”. We change “was applied in the experiment” into “were applied in the experiment”

11. Line 84， “pH of nutrient” should be “The pH of nutrient”. We change “pH of nutrient” into “The pH of nutrient”

12. Line 104-106， “The classification of H values is” should be “classifications of H values are”. We change “The classification of H values is” into “classifications of H values are”

13. Line 106-107, “it often is” should be “it is often”. We change “it often is” into “it is often”

14. Line 106-107, “fit” should be “fitting”. We change “fit” into “fitting”

15. Line 146， Fig 1 “supply ” should be “supplies”. We change “supply” into “supplies”

16. Line 207-209，“decreased in plant subjected high” should be “decreased in plants subjected to high”. We change “decreased in plant subjected high” into “decreased in plants subjected to high”

17. Line 212, “Comparing with” should be “Compared with”. We change “Comparing with” into “Compared with”

18. Line 214, “N:P ratio in leaf of was” should be “N:P ratio in leaf was”. We change “N:P ratio in leaf of was” into “N:P ratio in leaf was”.

19. Line 215-216, “more studies are need” should be “more studies are needed”. We change “more studies are need” into “more studies are needed”

20. Line 215-216, “between N:P ration” should be “between N:P ratio”. We change “between N:P ration” into “between N:P ratio”

21. Line 220-221, “presented positive influence” should be “presented a positive influence”. We change “presented positive influence” into “presented a positive influence”

22. Line 224, “growth rate of plant and its leaf N:P ratio” should be “growth rate of plants and the N:P ratio”. We change “growth rate of plant and its leaf N:P ratio” into “growth rate of plants and the N:P ratio”

23. Line 230, “nutrient supply” should be “nutrients supply”. We change “nutrient supply” into “nutrients supply”

24. Line 234, “Jinagxi Province” should be “Jiangxi”. We change “Jinagxi Province” into “Jiangxi”

25. Line 240， “ may be play” should be “ may play” or “maybe play”. We change “may be play” into “maybe play” 

Reviewer#2:

1. Line6: “N1:1 mmol L-1”. The expression is not clear on what is added. Details on the nitrogen or phosphorus supply were mentioned in the sentence (line83-85).

2. Line 7-9: Throughout the manuscript, author refer to nutrient ‘contents’. I think it would be more accurate to refer to nutrient ‘concentrations’ in the plant tissue. We change ‘contents’ into ‘concentrations’

3. Line 14: Local abundance of species was not studied and the grammar of this sentence has error. Significant correlation between invasion ability and relative growth rate was not observed in the three alien plants. So, it is suggested that local abundance of species, human activities, invasive history et al may play an important role in invasion of alien plants as well as relative growth rate. 

4. Line 19-20: There are many other mechanisms by which plants adapt to nitrogen and phosphorus deficient soils. It would be useful to briefly discuss these before focusing on leaf stoichiometry. N:P ratio is a critical indicator of nutrient limitation (N vs P) in the terrestrial ecosystem. Leaf stoichiometry can reflect nutrient allocation strategy, growth strategy and expanding ability of invasive plants.

5. Line 24-25：In introduction, leaf stoichiometry would be changed by different nitrogen and phosphorus supply. The pattern is inconsistent among different plants. It is suggested that author may reorganize the sentence to express clearly. We rewritten the sentence into “Different nitrogen and phosphorus supply bring about changes of leaf stoichiometry, and these changes are various among different plants”. 

6. Line 48-49: The sentence is difficult to understand. It is suggested that author rewrite the sentence to express clearly. We rewritten the sentence into “However, the clear relationship between nutrient absorption capacity of alien plants and their expanding ability was not established in other studies”.

7. In the section of materials and methods, further detail on the three species is not enough. It is suggested that more details on the three species. More information on the three alien plants was supplemented in line54-56 and line237-242.

8. In the section of tables and figures, the units for relative growth rate (i.e. g g–1 d–1) should be g day–1 (growth per day)? We have made correction according to the comments.

9. Line 2, change “change” into “changes”. We change “change” into “changes”

10. Line 7-8, change “leaf” into “leaves”. We change “leaf” into “leaves”

11. Line 10, change “level” into “levels”. We change “level” into “levels”.

12. Line 40, change “……, positive correlation between……” into “……, a positive correlation between……”. We change “……, positive correlation between……” into “……, a positive correlation between……”.

13. Line 43, change “……between N:P ratio of leaf and…..” into “…..between N:P ratios of leaves and…..”. We change “……between N:P ratio of leaf and…..” into “…..between N:P ratios of leaves and…..”.

14. Line 45, change “Invasion of alien plants is a severe threat to biodiversity and ecosystem worldwide.” into “The invasion of alien plants severely threatens biodiversity and ecosystem worldwide.” We change “Invasion of alien plants is a severe threat to biodiversity and ecosystem worldwide.” into “The invasion of alien plants severely threatens biodiversity and ecosystem worldwide.”

15. Line 49，change “bewteen” into “between”. We change “bewteen” into “between”.

16. Line 65, change “stolen” into “stolon”. We change “stolen” into “stolon”.

17. Line 67, change “Each node along stolon of…..” into “Each node along the stolon of…..”. We change “Each node along stolon of…..” into “Each node along the stolon of…..”.

18. Line 81，change “was applied in the experiment” into “were applied in the experiment” We change “was applied in the experiment” into “were applied in the experiment”

19. Line 158, change “diffdifferences” into “differences”. We change “diffdifferences” into “differences”

20. Line 158, change “Table 2 and 4” into “Tables 2 and 4”. We change “Table 2 and 4” into “Tables 2 and 4”.

21. Line 234, change “Jinagxi Province” into “Jiangxi”. We change “Jinagxi Province” into “Jiangxi”.

We greatly appreciate help from you and two anonymous referees again. We hope that the revised manuscript may be acceptable for publication. We look forward to hearing from you.

Sincerely

 Dongwei Yu

---

## [Editor Report · Decision Letter 2]

22 Nov 2022

Effects of nutrient supply on leaf stoichiometry and relative growth rate of three stoloniferous alien plants

PONE-D-22-22959R2

Dear Dr. Chen,

We’re pleased to inform you that your manuscript has been judged scientifically suitable for publication and will be formally accepted for publication once it meets all outstanding technical requirements.

Kind regards,

Xiao Guo, Ph.D.

Academic Editor

PLOS ONE

---

## [Editor Report · Acceptance letter]

25 Nov 2022

PONE-D-22-22959R2 

Effects of nutrient supply on leaf stoichiometry and relative growth rate of three stoloniferous alien plants 

Dear Dr. Chen:

I'm pleased to inform you that your manuscript has been deemed suitable for publication in PLOS ONE. Congratulations! Your manuscript is now with our production department. 

Kind regards, 

on behalf of

Dr. Xiao Guo 

Academic Editor

PLOS ONE